# Virus-borne mini-CRISPR arrays are involved in interviral conflicts

Sofia Medvedeva[1,2,3], Ying Liu[1], Eugene V. Koonin [4], Konstantin Severinov[2,5,6], David Prangishvili[1,7] & Mart Krupovic [1*]

CRISPR-Cas immunity is at the forefront of antivirus defense in bacteria and archaea and specifically targets viruses carrying protospacers matching the spacers catalogued in the CRISPR arrays. Here, we perform deep sequencing of the CRISPRome—all spacers contained in a microbiome—associated with hyperthermophilic archaea of the order Sulfolobales recovered directly from an environmental sample and from enrichment cultures established in the laboratory. The 25 million CRISPR spacers sequenced from a single sampling site dwarf the diversity of spacers from all available Sulfolobales isolates and display complex temporal dynamics. Comparison of closely related virus strains shows that CRISPR targeting drives virus genome evolution. Furthermore, we show that some archaeal viruses carry mini-CRISPR arrays with 1–2 spacers and preceded by leader sequences but devoid of *cas* genes. Closely related viruses present in the same population carry spacers against each other. Targeting by these virus-borne spacers represents a distinct mechanism of heterotypic superinfection exclusion and appears to promote archaeal virus speciation.

[1] Institut Pasteur, Department of Microbiology, 75015 Paris, France. [2] Center of Life Sciences, Skolkovo Institute of Science and Technology, Skolkovo, Russia. [3] Sorbonne Université, Collège doctoral, 75005 Paris, France. [4] National Center for Biotechnology Information, National Library of Medicine, Bethesda, MD 20894, USA. [5] Waksman Institute, Rutgers University, Piscataway, NJ 08854, USA. [6] Institute of Molecular Genetics, Moscow 123182, Russia. [7] Ivane Javakhishvili Tbilisi State University, Tbilisi 0179, Georgia. *email: mart.krupovic@pasteur.fr

The incessant struggle between viruses and cells drives the evolution of both conflicting parties, structuring their populations across time and space, spawning major evolutionary innovations, and affecting the major biogeochemical cycles on our planet[1–5]. At the interface of virus-host interactions, various mechanisms of defense and counter-defense have emerged[6–10]. These vary from physical barriers, which abrogate the delivery of foreign genetic material into the host interior, to specific recognition and degradation of the invading nucleic acids inside the cell, to suicide of infected cells that can save the clonal population[11,12]. Concurrently, viruses and other mobile genetic elements (MGEs) evolved elaborate ways to overcome the host defenses. The prime example of such systems in many bacteria and most archaea is the CRISPR-Cas adaptive immunity and the MGE-encoded anti-CRISPR proteins[8,13]. The defense systems evolve by widely different mechanisms which often involve recruitment of MGEs or their components[14–16]. Once in existence, the defense and counter-defense systems can change their 'owner' according to the 'guns-for-hire' concept[17,18]. Indeed, CRISPR-Cas systems are not exclusive to cellular organisms and have been captured and exploited by various MGEs, including bacteriophages, plasmids, and transposons[19–23].

The functioning of CRISPR-Cas system can be divided into three major stages: (i) adaptation stage, during which Cas1-Cas2 complex, in some variants including additional Cas proteins (also called the adaptation module), incorporates new virus/plasmid-derived sequences (spacers) into a CRISPR array, next to the promoter-containing leader sequence; (ii) processing stage, during which the CRISPR array is transcribed and the transcript is processed into separate CRISPR (cr) RNAs containing the spacer sequence with 5′ and 3′ tags derived from the flanking repeats; (iii) interference stage, during which the crRNA binds the CRISPR effector complex (also called interference module) which recognizes and degrades DNA and/or RNA molecules of encountered MGEs[13,18,24–28]. Based on fundamental differences in the organization of the effector modules and the presence of unique signature genes, CRISPR-Cas systems are classified into two classes that include six types and numerous subtypes[13,29].

Hyperthermophilic archaea of the order Sulfolobales harbor some of the most complex among the studied CRISPR-Cas systems: most of the genomes contain several CRISPR arrays with different CRISPR repeats, several adaptation modules and several type I and type III interference modules[30]. Concurrently, members of the Sulfolobales harbor an extremely diverse virome[31–33]. As a countermeasure to sophisticated defense systems of the host, at least some viruses of Sulfolobales encode anti-CRISPR proteins[34,35]. CRISPR-Cas immunity of Saccharolobus (until recently known as Sulfolobus) has been extensively explored in vitro, providing insights into the mechanisms of adaptation, expression and interference[36–38]. In parallel, in vivo experiments have demonstrated that new spacers can be inserted into the CRISPR arrays upon infection with a single or multiple viruses[39,40]. Interference with the targeted MGE at the level of DNA and/or RNA has been described for different CRISPR interference modules[41,42].

The sequence of each CRISPR spacer and its position in the array, respectively, provide information on the encountered MGEs and the order of their interaction with the host. Analysis of the CRISPR spacer content in Saccharolobus populations demonstrated high spacer diversity[43,44], reassortments of CRISPR arrays between strains[45], as well as specificity of CRISPR spacers to local viruses[46,47]. Here we perform deep sequencing of the CRISPRome—all spacers contained in a microbiome—of hyperthermophilic archaea of the order Sulfolobales recovered directly from environmental samples and from laboratory enrichment cultures. Analysis of the spacer sets reveals biogeographical pattern in viral communities and complex temporal dynamics of CRISPR spacers. We discover that some of the most abundant spacers in the CRISPRome come from mini-arrays carried by archaeal viruses themselves. Spacers from these mini-arrays target closely related viruses present in the same population and likely mediate CRISPR-dependent superinfection exclusion and promote archaeal virus speciation.

## Results

**Massive diversity of environmental CRISPR spacers.** To gain insights into the diversity and dynamics of the CRISPRome, we studied the natural population of Sulfolobales in the previously characterized environmental samples from a thermal field in Beppu, Japan[48] (see "Methods"). To this end, we used CRISPR repeat-specific primers to amplified by PCR[49] the CRISPR spacers associated with the four principal CRISPR repeat sequences present in Sulfolobales[50], followed by high-throughput sequencing (HTS) of the amplicons (see Methods). Notably, in Saccharolobus, the interference modules of types I and III can utilize crRNA from CRISPR arrays with different repeat sequences[51], precluding unambiguous assignment of CRISPR arrays to interference modules. Thus, hereinafter, we refer to the four consensus CRISPR repeat sequences as A, B, C, and D (Fig. 1a). All four CRISPR repeat sequences are exclusive to the Sulfolobales, including the genera Sulfolobus, Saccharolobus, Acidianus and Metallosphaera (Supplementary Table 1). The temporal dynamics of the CRISPRome was analyzed in two parallel series of enrichment cultures established from environmental samples J14 and J15 (ref. [48]), in media that favor the growth of Sulfolobus/Saccharolobus and Acidianus species prevalent in terrestrial hot springs and that grow well under laboratory conditions (Supplementary Fig. 1).

More than 25 million spacers were sequenced from all the samples (Supplementary Table 2), which after clustering of sequences with 85% identity resulted in 40,704 unique spacer clusters (Supplementary Data 1). The clustered spacer collection obtained here from a single sampling site dwarfs (sixfold increase) the size of the Sulfolobales spacer database from strains ($n = 6699$ unique spacers) that have been previously isolated from geographically diverse locations (Fig. 1b). The largest intersection (48 unique spacers) was found between the obtained spacer set from Beppu and spacers from the previously sequenced Sulfolobales strains isolated in Japan (Fig. 1c), indicative of the presence of a biogeographical pattern in the Sulfolobales virome, consistent with previous observations from other geographical locations[46,47]. The original environmental sample comprised 86% of the 40,704 spacer clusters, with 64% of spacers found exclusively in this sample. In contrast, the 10-days and 20-days enrichment samples, respectively, contained only ~20% and ~15% of the total collection of Beppu spacers (Supplementary Fig. 2a). The massive loss of spacer diversity must result from extinction of certain Sulfolobales strains during cultivation under laboratory conditions. Indeed, we found that the initially less abundant spacers (with coverage < 30) were the most strongly affected by the cultivation procedure, with 85% disappearing in the enrichment cultures, whereas only 7% of initially dominant spacers, sequenced more than 500 times, were lost after 20 days (Supplementary Fig. 2b). This result indicates that, as one would expect, the bottleneck primarily affects the strains with a small population size.

Most amplicons including 2 or 3 spacer-repeat units and sequenced >100 times could be assembled into longer CRISPR arrays (>3 spacers) through identical spacers (Supplementary Text). The longest assembled CRISPR arrays were 131, 66, 139 and 119 for spacers associated with the CRISPR-A, -B, -C and -D repeats, respectively.

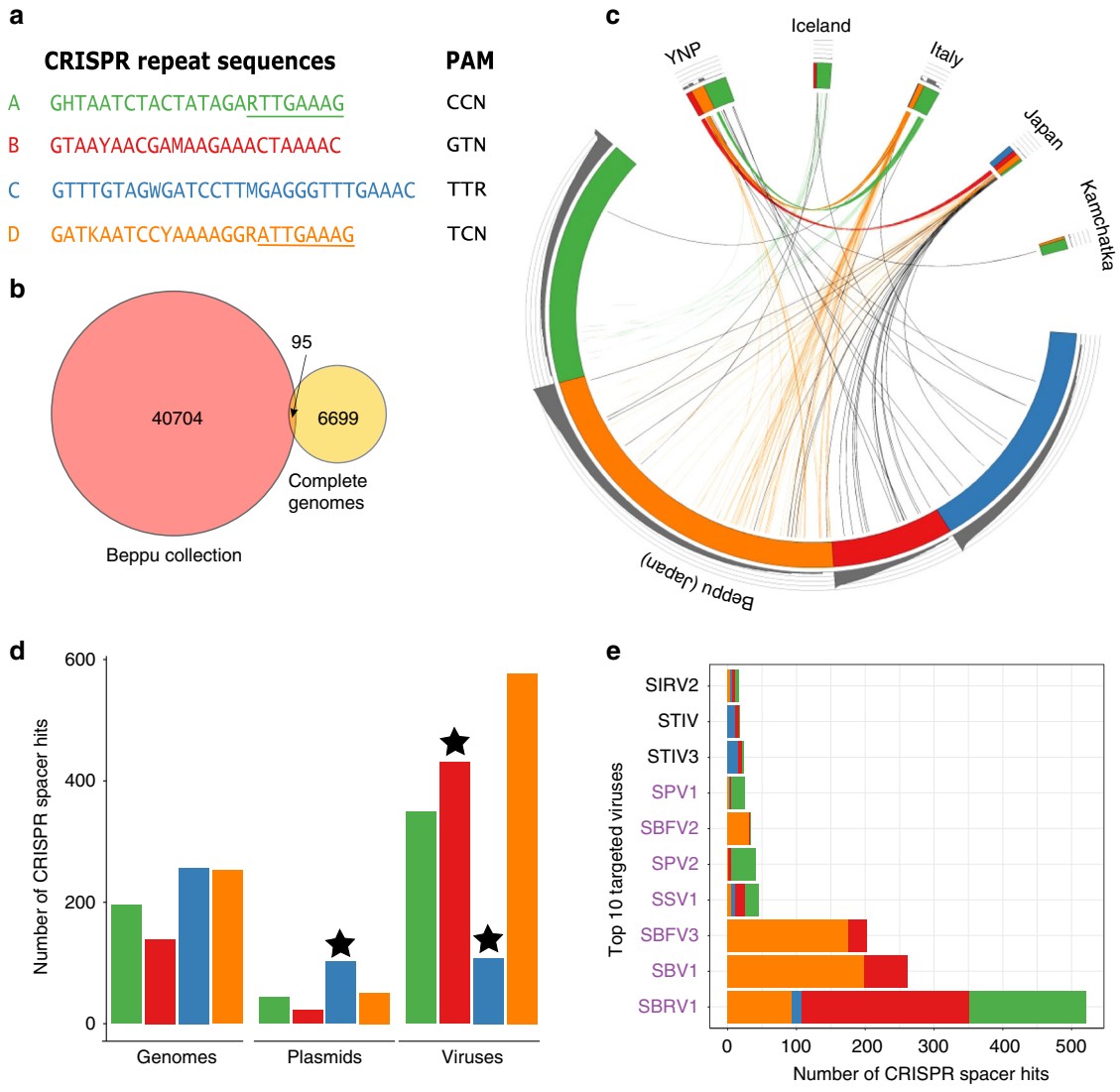

**Fig. 1** Characteristics of the analyzed spacer collections. **a** The four principal repeat sequences found in Sulfolobales; the color-coded repeat sequences and the corresponding PAMs are shown. The last eight nucleotides shared between CRISPR-A and CRISPR-D are underlined. IUPAC nucleotide code is used to show variations in nucleotide sequence of CRISPR repeats in different Sulfolobales genomes: H = A, C or T; Y = C or T; R = A or G; M = A or C; W = A or T; K = G or T. **b** Intersection of the Beppu spacer collection and spacers from sequenced Sulfolobales isolates available in public databases. The numbers represent the actual number of spacer clusters. **c** A circular diagram of spacers amplified from the Beppu hot spring and spacers from the Sulfolobales isolates clustered by the country of origin. Spacers belonging to arrays with the four principal repeat sequences found in Sulfolobales are indicated by identical colors matching those used in (**a**). Spacers that differ from each other by less than two nucleotides are connected by lines whose colors correspond to colors indicating CRISPR consensuses. Black lines connect spacers shared by arrays of different types. The outer gray histograms represent the abundance of CRISPR spacers in log10 scale. YNP, Yellowstone National Park, United States. **d** The bar plot showing the numbers of protospacers found in Sulfolobales host genomes, plasmids and viruses. The stars indicate values that differ significantly (chi square test, P-value < 0.001) from the expectation. Colors represent spacers associated with different CRISPR consensuses, as in (**a**). **e** The bar plot shows the numbers of protospacers found in the top 10 targeted Sulfolobales viruses. Names of viruses isolated in Beppu, Japan are highlighted with violet color. Source data for **c**, **d**, and **e** are provided as a Source Data file

**Provenance of the Sulfolobales spacers**. To assess the provenance of the spacers, we matched the Beppu spacer set against the available Sulfolobales genomes, viruses and plasmids (Fig. 1c). Using the threshold of >85% identity over the full length of the alignment, protospacers were identified for ~6% of spacers, a value that is close to the ~7% mean observed in previous analyses of the global dataset of spacers from all available sequenced genomes[52]. Unexpectedly, protospacers associated with the CRISPR-C array were overrepresented in plasmids (chi square test, P-value < $10^{-5}$) and underrepresented in viruses (chi square test, P-value < $10^{-36}$), suggesting specialization among the distinct CRISPR consensuses to combat different types of MGE. The CRISPRome of the Sulfolobales community from Beppu included spacers against 53 viral genomes isolated from all over the world, but the most frequently targeted ones were those sequenced from the same sampling site[48], further indicating local adaptation of the Sulfolobales viruses. Notably, fusellovirus SSV1 isolated from the same Beppu site 35 years ago[53] is the fourth most targeted virus, suggesting that SSV1 and its derivatives are persistent components of the Beppu virome (Fig. 1d). Spacers associated with different CRISPR repeat types showed specificity to different viruses (Fig. 1d, Supplementary Fig. 3), possibly

reflecting distinct host ranges of the corresponding viruses. For example, related viruses SBFV1 and SBFV3 are primarily targeted by spacers from CRISPR-A and CRISPR-D, respectively. Rudivirus SBRV1 is targeted by as many as 521 unique spacers belonging to different CRISPR consensuses, signifying that SBRV1-like viruses are or were associated with broadly diverse hosts. Such dense coverage of protospacers would allow reconstruction of 53% of the SBRV1 genome. Moreover, tiling the sequences of overlapping spacers allowed assembly of several additional viral contigs (see Supplementary Text; Supplementary Fig. 3c). Finally, mapping the spacers against the Sulfolobales chromosomes proved to be an efficient approach to identify integrated MGEs (see Supplementary Text; Supplementary Table 4).

**Temporal CRISPRome dynamics**. To explore the temporal CRISPRome dynamics, we compared the distributions of frequencies of spacers across samples, including the original environmental sample and the enrichment cultures grown in *Sulfolobus/Saccharolobus*- and *Acidianus*-favoring media (Fig. 2). Despite possible biases introduced by PCR amplification, CRISPR spacers sequenced from the same replicon generally get similar representation in HTS reads[49]. Therefore, the abundances of spacers show a multimodal distribution (Fig. 2), likely reflecting the number of spacer-carrying Sulfolobales strains in the sample. However, the possibility that the highly represented groups include spacers from more than one strain cannot be formally excluded. Comparison of the temporal variation in the spacer abundances revealed significant differences between the J14 and J15 samples (Supplementary Text). Given that the strains from both samples were propagated under the same conditions, and initially displayed similar spacer composition (Supplementary Fig. 4), we hypothesized that viruses present in enrichment cultures might be responsible for the differences in the growth dynamics of some of the strains. Indeed, we have previously shown that samples J14 and J15 contain different, albeit overlapping, virus populations[48]. Whereas J14 contains SBV1, SBFV1, SBFV3, SBRV1, and SPV2, J15 contains SBV1, SBFV1, SBFV2 and SPV1 (Supplementary Fig. 1). Among these, SPV1 and SPV2, icosahedral, non-lytic viruses of the family *Portogloboviridae*[54,55], are by far the most abundant in the respective samples, accounting for ~90% of all virome reads[48].

To understand the reasons underlying the dominance of SPV1 and SPV2 and their exclusivity to the corresponding samples, we focused on the comparison of spacers targeting the two viruses in J14 and J15, respectively. Notably, the genomes of SPV1 and SPV2 are 92% identical to each other[48] and mapping of the CRISPR spacers from our dataset showed that genomic location of sequence divergence between SPV1 and SPV2 specifically coincides with targeting by CRISPR spacers (chi square test, P-value < 0.01; Fig. 3a). Thus, CRISPR targeting is an important factor driving the genome evolution of portogloboviruses.

**Characterization of virus-borne mini-CRISPR arrays**. Six spacers associated with the CRISPR-A repeat and matching SPV1 and SPV2 were the most abundant in the corresponding enrichment cultures (Supplementary Data 2). Unexpectedly, 3 of these spacers matching SPV1 (100% identity) were sequenced from the J15 enrichment culture dominated by SPV1, and conversely, the 3 spacers matching SPV2 (100% identity) were sequenced from the J14 enrichment dominated by SPV2. Furthermore, abundant CRISPR spacers in our data could be assembled into long (>3 spacers) CRISPR arrays (see above and Supplementary Text). However, despite being among the most abundant in our dataset, the 6 SPV1- and SPV2-matching spacers

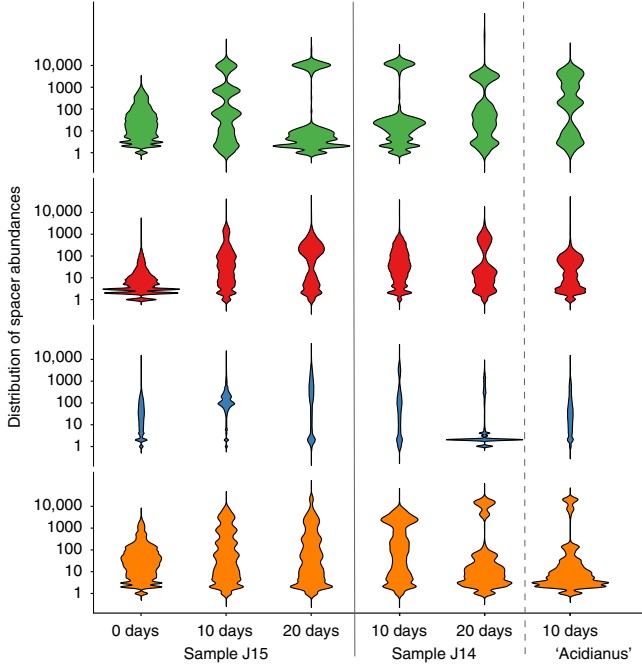

**Fig. 2** The violin plots show the density of the distribution of spacer abundances in the environmental sample and enrichment cultures established from samples J14 and J15. In the J14 sample, the enrichment culture established in the *Acidianus*-favoring medium is separated from those established in the *Sulfolobus/Saccharolobus*-favoring medium by a dashed line. Plots in each row represent spacers, associated with different CRISPR consensuses and color-coded as in Fig. 1a. Plots in each row are scaled to have the same area. Log10 scale for the abundance values was used. Source data are provided as a Source Data file

could not be reconstructed into long arrays, but instead appeared to be located within mini-CRISPR arrays each carrying 1 or 2 spacers.

To better understand the dominance of SPV-matching spacers, we analyzed the corresponding loci in the viral SPV1 and SPV2 genomes and, unexpectedly, found that the mini-CRISPR arrays including CRISPR-A repeats flanking the SPV-targeting spacers are encoded in intergenic regions of both SPV1 and SPV2 genomes. Thus, the 6 most abundant CRISPRome spacers were amplified from mini-CRISPR arrays in SPV1 and SPV2 genomes, rather than from the Sulfolobales genomes. This inference was validated experimentally by PCR amplification and sequencing of the mini-CRISPR arrays using as templates DNA extracted from purified SPV1 virions as well as from the 20-days enrichment cultures J14 and J15 used for CRISPRome sequencing (Supplementary Fig. 5). The results showed that the mini-CRISPR arrays sequenced in the course of CRISPRome analysis were indeed amplified from the SPV1 and SPV2 genomes present in the enrichment cultures. Thus, the dominance of the SPV1 and SPV2 viruses in the J14 and J15 cultures explains the abundance of the corresponding spacers in the corresponding samples.

The relative positions of the mini-CRISPR arrays containing two spacers in the SPV1 and SPV2 genomes were the same (Fig. 3b), but the corresponding spacers were different, implying active spacer turnover. These mini-CRISPR arrays are preceded by the promoter-containing leader sequences similar to those found in genomic *Saccharolobus* CRISPR arrays (Fig. 4). Unlike in the case of certain bacteriophages and integrated MGEs, which carry complete CRISPR-Cas systems[19,23], the SPV-encoded mini-CRISPR arrays are not accompanied by recognizable virus-

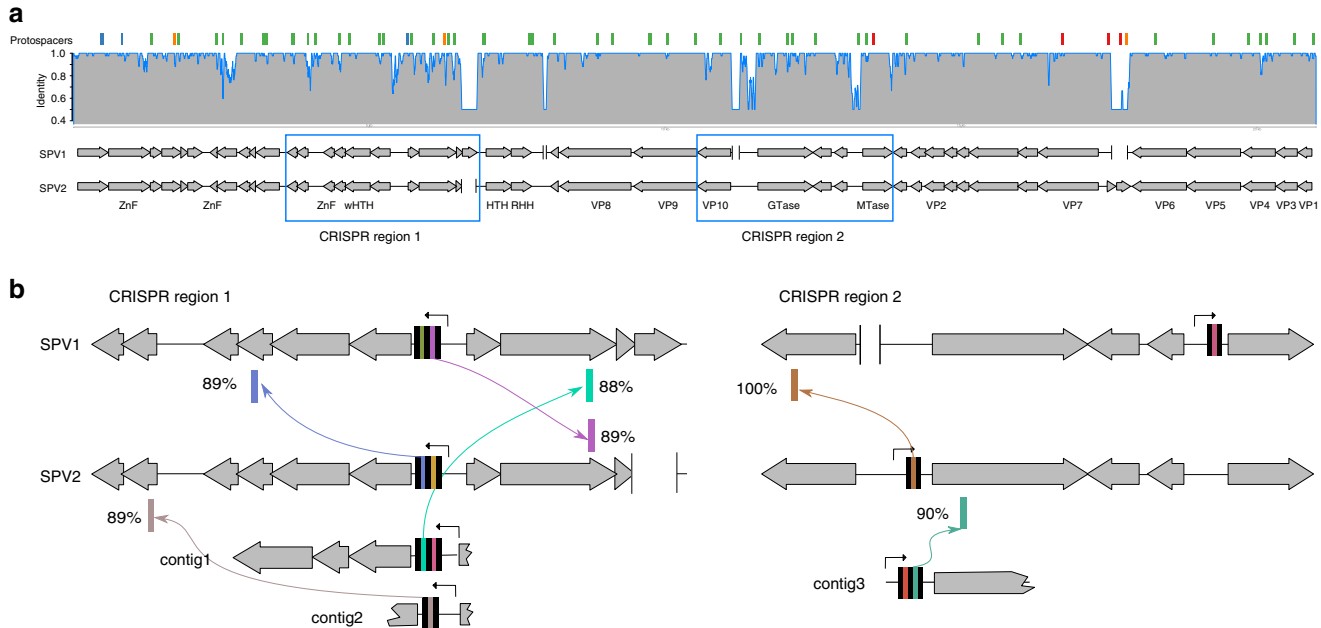

**Fig. 3** Mini-CRISPR arrays in SPV1 and SPV2 genomes. **a** Comparison of the SPV1 and SPV2 genomes. Genes are represented with arrows following the direction of transcription. Deletions in one of the two genomes with respect to the other are indicated as gaps bordered with vertical lines. Gray histogram above the genome maps shows the identity calculated in 50 bp window from the SPV1-SPV2 nucleotide alignment. Locations of protospacers are showed as colored bars at the top of the figure. The regions zoomed-in in (**b**) are boxed. ZnF, zinc finger protein; (w)HTH, (winged) helix-turn-helix domain-containing protein; RHH, ribbon helix helix domain-containing protein; MTase, methyltransferase; GTase, glycosyltransferase; VP, virion protein. Source data are provided as a Source Data file. **b** Zoom-in on two regions of the SPV1 and SPV2 genomes carrying mini-CRISPR arrays (CRISPR region 1 and CRISPR region 2). Black bars represent CRISPR repeat. The predicted promoters in the leader sequences are indicated with broken arrows. Positions of hits of spacers from mini-CRISPR arrays carried by SPV-like viruses are shown with colored bars and arrows link the spacers and the corresponding protospacers. Identities between spacer and protospacers are indicated next to the protospacer bars. Three mini-CRISPR arrays found in the virome data are shown below the corresponding regions of alignment and labeled as contigs 1 to 3

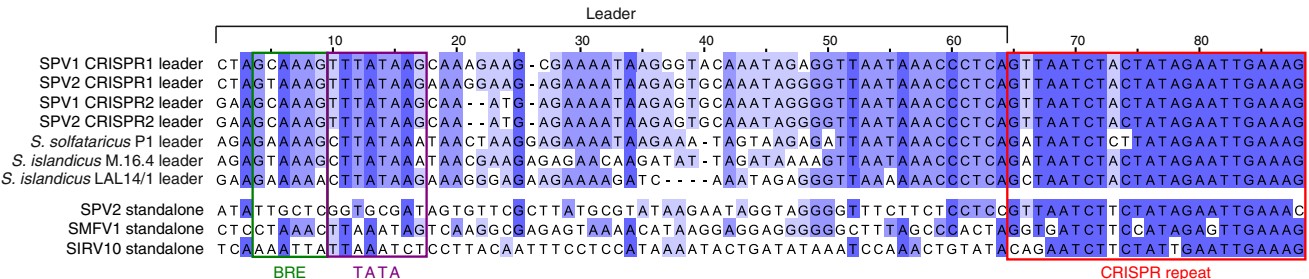

**Fig. 4** Alignment of the loci including the leader sequences and CRISPR repeats associated with the *Saccharolobus* CRISPR arrays and virus-borne mini-CRISPR arrays. The bottom 3 sequences correspond to stand-alone CRISPR repeats. BRE and TATA elements found in the promoters of the leader sequences are boxed

encoded *cas* genes. However, given the sequence similarity of the repeats and leader sequences to the corresponding elements of the host (Fig. 4)[38], it is highly likely that new spacers can be inserted by the endogenous host-encoded adaptation modules. Consistent with this possibility, some of the genetic tools designed for *Saccharolobus* specifically rely on the recruitment of endogenous Cas proteins to function with artificially designed, plasmid-borne CRISPR spacers targeting a gene of interest[42,56].

Remarkably, two of the three spacers carried by SPV2 target SPV1, whereas only one of the three spacers carried by SPV1 targets SPV2 (Fig. 3b), with another one targeting a pRN1-like plasmid integrated in the *S. tokodaii* genome (BA000023, nucleotide coordinates 328508–335407). Importantly, the loci orthologous to the regions targeted by spacers in the viruses carrying the spacers contain either point mutations or deletions,

preventing self-targeting (Supplementary Data 2). Notably, the SPV1 and SPV2 spacers target regions close to the mini-CRISPR arrays, although origins and consequences of this proximity remain unclear (Fig. 3b). These findings prompted us to search for additional mini-CRISPR arrays in our CRISPRome dataset, resulting in 15 more candidates (Fig. 5a). Three of the mini-CRISPR arrays were confirmed to be encoded within viral genomes by analysis of the previously sequenced viromes from J14 and J15 samples[48]. All three were found in the virome contigs[48] containing fragments of genes orthologous to those of SPV1 and SPV2 (Fig. 3b). We suggest that these additional mini-CRISPR arrays are carried by minor strains of SPV-like viruses present in the population. Of the 26 spacers carried by the 15 candidate mini-CRISPR arrays, 18 were found to target different loci within the SPV1 or SPV2 genomes (Fig. 5a).

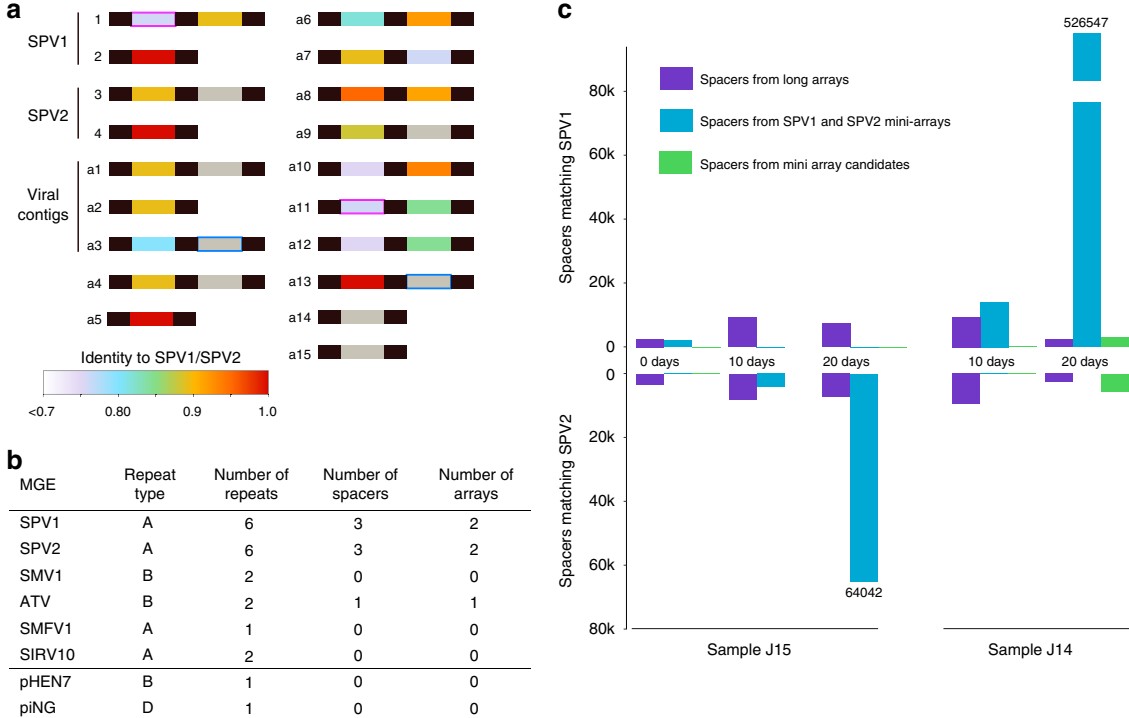

**Fig. 5** Mini-CRISPR arrays in viral genomes. **a** mini-CRISPR arrays predicted from the CRISPRome data (labeled as a1 through a15). Identity of spacers to SPV1 or SPV2 genomes is color-coded with the scale provided at the bottom of the figure. Mini-CRISPR arrays found in SPV1, SPV2 and 3 viral contigs shown in Fig. 3b are labeled on the left. Pink and blue boxes show two pairs of identical spacers. **b** Mini-CRISPR arrays and standalone repeats in Sulfolobales viruses and plasmids. **c** Total abundance of SPV1 and SPV2 matching spacers from long (cellular) CRISPR arrays and mini-CRISPR (viral) arrays from SPV1 and SPV2 as well as minor SPV strains. Source data are provided as a Source Data file

To assess the generality of the potential CRISPR-mediated virus-virus interactions, we searched if any of the other available genomes of Sulfolobales viruses and plasmids carry CRISPR repeats of the four consensuses. A mini-CRISPR array has been also identified in the genome of Acidianus two-tailed virus (family *Bicaudaviridae*)[57]. It consisted of a single spacer flanked by CRISPR-B repeats. In addition, stand-alone CRISPR repeats similar to those of the corresponding host species were identified in the genomes of SPV1 and SPV2 as well as three other archaeal viruses and two conjugative plasmids (Fig. 5b). However, the stand-alone repeats were not preceded by recognizable leader sequences (Fig. 4). Whether such repeats are competent targets for spacer integration is thus unclear.

To assess the effects of virus-mediated versus host-mediated CRISPR immunity against SPV1 and SPV2, we compared the number and abundance of the targeting spacers originating from the long host-borne CRISPR arrays and virus-borne mini-CRISPR arrays (Fig. 5c). Note that the abundance of spacers in the samples reflects the abundance of the organisms, viruses or cells, carrying the corresponding CRISPR arrays. In the initial environmental sample J15 and in 10-days enrichments, spacers from the long arrays were the major contributors to the total immunity against SPV1 and SPV2 viruses. However, after 20 days, the abundance of spacers from mini-arrays increased dramatically. Moreover, spacers from the host arrays targeted SPV1 and SPV2 indiscriminately (judging from the identity between spacers and protospacers), whereas spacers from mini-arrays were specific against either SPV1 or SPV2. Thus, SPV1 and SPV2 came to dominance in the 20-days cultures of J14 and J15 samples, respectively, consistent with the enrichment of the SPV1-borne and SPV2-borne spacers/mini-arrays in the corresponding samples. Notably, in the J14 20-days enrichment, we observed the rise of minor SPV variants with spacers against SPV1 and SPV2 (Fig. 5c).

## Discussion

Collectively, our results demonstrate the utility of the CRIS-PRome for understanding virus-host interactions and reveal a potential strategy used by viruses to restrict competing MGE via CRISPR-mediated superinfection exclusion. A recent, independent comparative genomic analysis of bacterial and archaeal viruses has demonstrated the presence of CRISPR mini-arrays and single-repeat units in many bacteriophage and prophage genomes as well as a few archaeal viruses[58]. Some of the spacers in the identified mini-arrays targeted adjacent genes in closely related virus genomes but not the mini-array-carrying virus itself, in full agreement with the pattern identified in the present work. However, unlike the case of SPV1 and SPV2 described here, these phage mini-arrays lack the leader regions, suggesting that they might acquire spacers via recombination with host arrays rather than canonical adaptation. Whether these spacer-repeat units are equivalent to the leader-less stand-alone CRISPR repeats identified in several archaeal viruses and plasmids described here (Fig. 5b) remains unclear and deserves further investigation.

The reciprocal CRISPR targeting by SPV1 and SPV2 strongly suggests that the virus-encoded mini-CRISPR arrays are involved in interviral conflicts and represent a distinct mechanism of heterotypic superinfection exclusion, whereby a cell infected by one virus becomes resistant to the other virus (Fig. 6). This possibility is consistent with the observation that the cultures contain either SPV2 (J14) or SPV1 (J15), not both (Supplementary Fig. 1). Furthermore, we hypothesize that avoidance of self-targeting promotes speciation in the portoglobovirus population. In a similar fashion, it has been recently suggested that CRISPR

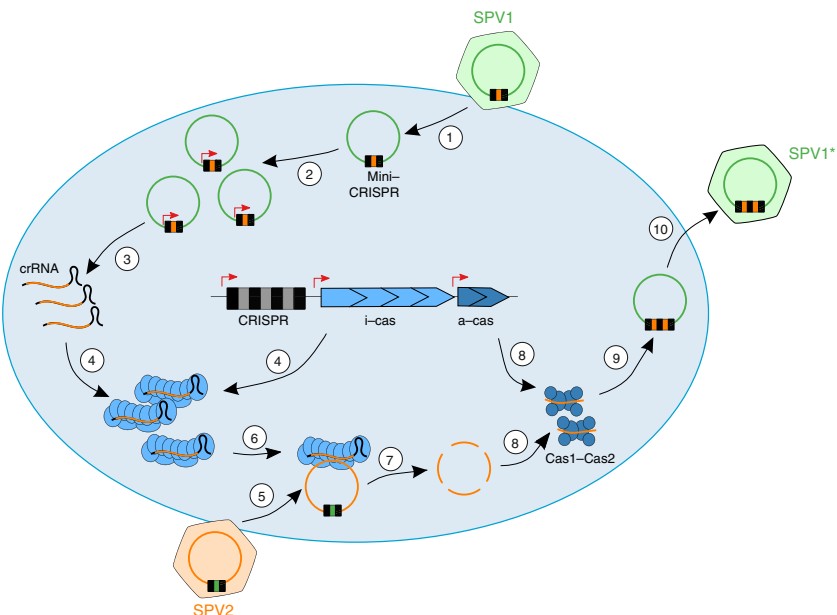

**Fig. 6** Proposed mechanism of CRISPR-mediated interviral conflict. 1: SPV1 carrying mini-CRISPR array with a spacer against SPV2 infects a susceptible host; 2: SPV1 genome replication results in increased copy number of the mini-CRISPR arrays compared to the chromosomal CRISPR arrays; 3: transcription of the mini-CRISPR arrays; 4: viral crRNAs are complexed with the cellular interference module (i-cas); 5: secondary coinfection of the same cell with SPV2; 6 and 7: SPV2 genome is recognized and degraded by the effector module; 8: fragments of the SPV2 genome are loaded on the cellular Cas1-Cas2 adaptation complex (a-cas); 9: mini-CRISPR array of SPV1 is upgraded with new spacers against SPV2; 10: New variant of SPV1 (SPV1*) is released and spreads the immunity in the population

spacers acquired during inter-species mating of halophilic archaea also influence speciation[59]. Importantly, SPV1 is a non-lytic virus, which establishes a chronic infection and is released without killing its host[54]. Therefore, the association between a non-lytic, CRISPR-bearing virus and the host is potentially beneficial to both parties and can thus be considered a form of symbiosis.

Interviral conflicts via virus-targeting mini-CRISPR arrays and other similar strategies are likely to contribute to viral genome evolution and speciation, and further validate the 'guns for hire' concept[18] under which components of various defense and counter-defense systems are commonly exchanged between viruses and their cellular hosts.

## Methods

**Description of samples**. The 20-days enrichment cultures established from two environmental samples, J14 and J15, were dominated by members of the genus *Saccharolobus* (85% in J14 and 79% in J15), unclassified members of the family Sulfolobaceae (14% in J14 and 20% in J15), and genera *Sulfurisphaera* and *Acidianus* (1% in both samples)[48]. The viral component of the enrichments included populations of seven viruses belonging to four different families[48]. Two of the viruses could be isolated and grown in pure cultures[54,60]. We were able to perform PCR amplification of the CRISPR spacers with the DNA extracted directly from the J15 sample, but not from the J14 sample, possibly, due to the insufficient biomass in the latter. The cultures propagated in the *Sulfolobus/Saccharolobus*-favoring medium displayed efficient growth of the biomass, whereas those propagated in the *Acidianus*-favoring medium grew poorly and were discontinued after 10 days of incubation. Thus, in total, we analyzed one environmental sample and five enrichments: two 10-days enrichments and two 20-days enrichments in *Sulfolobus/ Saccharolobus*-favoring medium, and one 10-days enrichment in the *Acidianus*-favoring medium.

**CRISPRome amplification**. To amplify CRISPR arrays of Sulfolobales from total DNA samples, six pairs of primers (Supplementary Table 3) were designed. Two pairs of primers, C1 and C2, were designed to cover the diversity of the CRISPR consensus C. The forward and reverse primers in each pair were complementary to the CRISPR repeats, with the forward and reverse primers partially overlapping in their 5′-termini. Amplification reactions were carried out with DreamTaq DNA polymerase (Thermo Fisher Scientific, UK) under the following conditions: initial

denaturation for 5 min at 95 °C, followed by 40 cycles of 30 s at 95 °C, 30 s at 42–53 °C (depending on the Tm of specific primers), and 60 s at 72 °C, and a final extension at 72 °C for additional 2 min. For each DNA sample with each primer pair, five individual PCR reactions were set up. No amplification was obtained with the primer pair G1, which was designed for CRISPR repeat ATTACTTTTCTCT TATGAGACTAGTAC predicted for a single CRISPR array with only 3 spacers in *S. tokodaii* 7 genome. Notably, the latter CRISPR array is not localized near *cas* genes in *S. tokodaii*. After amplification, individual reactions were pooled and processed jointly. Amplicons of 100–300 bp in length were purified from the 1% agarose gel and sequenced on MiSeq (Illumina) with paired-end 250-bp read lengths (Genomics Platform, Institut Pasteur, France).

**Evaluation of primer specificity**. To verify specificity of our primers for each CRISPR repeat type, we calculated the fraction of spacers that could be found between CRISPR repeats of different types, allowing two mismatches and end gaps/ insertions. For CRISPR-A and -D repeats, we obtained 1.5 and 2% of intersection (Supplementary Fig. 6), i.e., 1.5% of spacers bordered by CRISPR-A repeats (and amplified with A-specific primers) were also found in arrays bordered with CRISPR-D repeats. Importantly, this fraction was comparable to those for other repeat types. For instance, for CRISPR-A and -C, the intersection was 2.8% and 2.6%, suggesting that there is no preferential cross-amplification of spacers associated with CRISPR-A and CRISPR-D repeats.

The association of homologous spacers with different CRISPR repeat consensuses can be explained by independent acquisition of the corresponding spacers from the same loci of the plasmid/viral genome by two distinct adaptation modules. This possibility was confirmed by analysis of spacer dataset from fully sequenced *Sulfolobus islandicus* genomes. With the same comparison parameters (two mismatches and end gaps/insertions), we found 14 out of 552 (2.5%) spacers intersecting between CRISPR-A and CRISPR-D repeats in fully sequenced genomes. This fraction closely matches that obtained in the analysis of the spacers in the Japanese Beppu dataset, and further suggests that cross-amplification of spacers associated with CRISPR-A and CRISPR-D repeats is negligible.

**Amplification of SPV-borne mini-CRISPR arrays**. Three pairs of primers were designed to amplify the SPV-borne mini-CRSPR arrays and their leader sequences (Supplementary Fig. 5; Supplementary Table 3). All primer pairs match perfectly to both SPV1 and SPV2 genomes. Amplification with the three pairs of primers using DNA isolated from SPV1 virions, J15 (20 days) and J14 (20 days) enrichment cultures yielded PCR products with expected sizes (Supplementary Fig. 5). The sequences of all nine amplified products were confirmed by Sanger sequencing.

**Spacer extraction and clustering**. Spacer sequences were extracted using spget program (https://github.com/zzaheridor/spget). Spget identifies degenerate sequences of CRISPR repeats and PCR primers, and extracts all sequences between them. To account for possible sequencing mistakes and natural CRISPR repeat diversity, additional 2–5 mismatches were allowed in repeat and primer sequences. Based on expected spacer lengths, extracted spacers shorter than 25 nt or longer than 60 nt were filtered out. An additional quality filter was applied—only spacers with all nucleotides sequenced with the Phred score value higher than 20 were used for further analysis. The described filtering resulted in the removal of ~25% of all spacers.

The clustering was performed by UCLUST program[61], with 85% identity threshold and zero penalties for end gaps. UCLUST algorithm was also used with 85% identity threshold to find common spacers for different sets. The coverage of spacer diversity was estimated with Good's criterion: $C = 1 – (N/\text{total number of clusters})$, where $N$ is the number of sequences that occurred only once or twice. The alpha diversity (Shannon entropy) and Chao estimate of coverage were calculated using the R package *vegan* (https://cran.r-project.org/web/packages/vegan/index.html). The spacer sequences are available in Supplementary Data 1.

**Reconstruction of CRISPR arrays**. The procedure of CRISPR array reconstruction uses pairs of neighboring spacers obtained from NGS reads. All pairs for the sample are joined into a directed graph, where each node represents a spacer, edges connect the spacers that appeared together in a pair, and the number of found pairs in NGS reads is used as an edge weight. The PCR amplification procedure could possibly lead to the emergence of chimeric pairs, when two independent spacers from different CRISPR arrays are falsely connected into a pair. For example, when an amplification product from one cycle (a primer-spacer-primer unit) is used as a long primer with 5′ overhang for the next cycle. Assuming that chimeric pairs are rare PCR artifacts, we filtered edges in our graph based on the weight. For each edge (pair of neighboring spacers), we calculate the sum weight of outgoing edges from the first spacer in the pair and the sum weight of incoming edges for the second spacer in the pair. If the weight of tested edge was lower than 5% of the calculated sums, the tested edge was removed (see Supplementary Fig. 7A). One example of reconstructed graph is shown in Supplementary Fig. 8. Several arrays share the same terminal, leader-distant spacer, some of the arrays are branching towards the leader end. The script for reconstruction of the CRISPR array graphs is implemented in R language (https://www.R-project.org/)[62]. Because this approach is not suitable for identification of mini-CRISPR arrays, we used the eccentricity metrics (the length of the longest path, which is going through the selected node). The eccentricity number shows the length of the longest CRISPR array, which can be reconstructed using selected spacer (see Supplementary Fig. 7B). Each Sulfolobales genome usually contains more than one CRISPR array with the same CRISPR repeat sequences. We observed groups of spacers from three independent graph components with linearly correlated frequencies in two samples (Supplementary Fig. 9), which is consistent with them being sequenced from the same genome.

**Protospacer analysis**. Protospacers were searched for with BLASTN[63] (word size 8, *e*-value < 0.01) in local databases of Sulfolobales viruses, plasmids and cellular genomes. PAMs were identified by aligning flanking sequences of protospacers. For Fig. 2A, chi-square test followed by Bonferroni correction was used to test the specificity of spacers associated with different CRISPR consensuses to different sources of protospacers (Sulfolobales host genomes, viruses or plasmids) based on total number of spacers for each CRISPR consensus and total number of hits to a particular source.

To calculate the *P*-value for the coincidence of spacer targeting and regions of divergence in SPV1 and SPV2 genomes, we aligned SPV1 and SPV2 genomes with ClustalW[64] and manually verified the quality of nucleotide sequence alignment. Next, we calculated the number of spacers in our dataset that match fully conserved regions of the SPV1-SPV2 alignment (22 spacers) and variable parts of the alignment (42 spacers). We simulated 10,000 spacers from random positions of SPV1-SPV2 alignment and calculated the number of simulated spacers from conserved regions (5674 spacers) and variable regions of the alignment (4326 spacers). Chi square test was performed for observed (22 and 42) and expected (5674 and 4326) values; the resulting *P*-value was 0.006.

**Loss of minor spacers during cultivation**. The error bars indicate the confidence interval for the proportion of lost spacers calculated as $\text{conf} = z(0.975)*\text{sqrt}(\text{lost}*(1-\text{lost})/N)$, where 'lost' is a fraction of lost spacers and $N$ is the total number of spacers in each group.

**Assembly of viral contigs from spacers**. To reconstruct the viral contigs, we performed "all spacers against all spacers" BLASTN (word size 8, identity > 0.7). Then a graph of spacers was built, where spacers are connected if they were matched by BLAST search. The graph was decomposed, spacers from the largest subgraphs were aligned with MUSCLE[65], and the alignment was manually corrected.

**Prediction of mini-CRISPR arrays in the CRISPRome data**. Mini-CRISPR arrays in the CRISPRome data were predicted based on the abundance of spacers. This section explains the abundance thresholds selected to distinguish spacers derived from long arrays and mini-arrays. By definition, mini-CRISPR arrays contain only 1 or 2 spacers. The procedure of CRISPR array reconstruction from the CRISPRome data resulted in multiple mini-CRISPR array candidates with only 1 or 2 spacers. For each of the candidates, we tested the alternative hypothesis (the array is a complete mini-array) versus null hypothesis (the array is incomplete and it is part of long array).

First, we analyzed mini-CRISPR array candidates with only one spacer. We calculated the frequency of sequencing reads with two spacers in each sample and estimate the probability ($p$) of spacer to be in the pair (~0.5, for J15 sample, 0-days enrichment) and corresponding probability to be alone ($1-p = 0.5$). Assuming that all spacers are independent, we calculated the probability to observe no pairs for the spacer, which was sequenced $N$ times: $P\text{-value} = (1-p)^N$. For the spacer sequenced 100 times, *P*-value was <0.01, so we defined a threshold of abundance $n = 100$ for the mini-CRISPR array candidates with only one spacer.

Second, we analyzed mini-CRISPR array candidates with two spacers. The probability for a spacer to appear as first spacer in the pair is 0.5. If the spacer was sequenced in the pair N times, 1 pair defines spacer as a first, and remaining $(N-1)$ pairs are used to estimate probability $0.5^{N-1}$ to observe spacer only as a first spacer in the pair. The same probability $0.5^{N-1}$ is to observe second spacer in the pair only as a second spacer. Then, $1-0.5^{N-1}$ is the probability that CRISPR array is incomplete from one side. And $(1-0.5^{N-1})^2$ is the probability that CRISPR array is incomplete from any of sides. Finally, the reverse value $1-(1-0.5^{N-1})^2$ is a *P*-value that two-spacer array is complete. With $n = 20$, *P*-value < 0.01; thus, we selected threshold $n = 20$ for mini-CRISPR array candidates with two spacers.

**Determination of the integration sites**. The precise boundaries of MGE integration were defined based on the presence of direct repeats corresponding to attachment sites (Supplementary Table 4). The direct repeats were searched for using Unipro UGENE[66].

**Reporting summary**. Further information on research design is available in the Nature Research Reporting Summary linked to this article.

## Data availability
The authors declare that the data supporting the findings of this study are available within the paper and its supplementary information files. The source data underlying Figs. 1c–e, 2, 3a, 5c and Supplementary Figs. S2, S3, S4, S6 and S9 are provided as a Source Data file.

## Code availability
Code used in this work is available from the authors upon request.

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

## Acknowledgements

This work was supported by l'Agence Nationale de la Recherche (project ENVIRA, #ANR-17-CE15-0005-01) and the European Union's Horizon 2020 research and innovation program under grant agreement 685778 (project VIRUS-X). E.V.K. is supported through the intramural program of the U.S. National Institutes of Health. S.M. was supported by Vernadski fellowship from Campus France, RSF 19-14-00323 and NIH grant GM10407 to K.S.

## Author contributions

S.M., K.S., D.P., and M.K. conceived the study. S.M. performed all sequence analyses. Y.L. amplified and sequenced the CRISPRome. S.M., E.V.K., K.S., and M.K. interpreted the results. S.M. and M.K. wrote the first draft of the manuscript. All authors edited and approved the final version of the manuscript.

## Competing interests

The authors declare no competing interests.
