## [Peer Review File · Nature Communications]

Reviewers' comments:

Reviewer #1 (Remarks to the Author):

Sofia et al. analyzed the CRISPRome of the hyperthermophilic archaea in two environmental samples prior to or after enrichment. They observed interesting phenomena indicative of CRISPR specification, local adaptation, and the interaction between CRISPRs and viruses. Especially, they investigated the virus-borne mini-CRISPRs carried by closely related viruses, and proposed their role in promoting virus speciation, principally based on the observation that one virus and spacers discriminatively targeting the other were coincidentally enriched in a sample. These findings are of great interest and ecological significance, but some data interpretation was arbitrary. Besides, some statements were confusing and conflicting, and several Figures were incorrectly cited (see below).

Major points:

- 1) Are the type-specific primers (A, B, C, D) good enough to distinguish the four CRISPR types? Especially for that the CRISPR repeat types A and D share the last a few nucleotides. The evidences for this specificity should be provided before counting the spacers belonging to the arrays with the four repeat sequences (Figure 1)
- 2) The estimated PCR sizes are likely incorrect. As the theoretical size of a PCR product containing one spacer is the sum of $(24-31) \times 2$ (size of two repeats), 40 (the average size of spacer), and 33×2 (two adaptors), which is around 165 bp. And this size will be around 235 bp for two-spacer containing PCR products. These sizes are out of the range (300-1000 bp) during PCR products purification (P9 Line301), but the authors implied that these products (Approximately 50% of our HTS sequencing reads include not solitary spacers but small fragments of CRISPR arrays with 2...) were the majority in the reads (P20 Line475).
- 3) P6 Lines 175-180: The 3 most abundant spacers targeting SPV1 were sequenced from J15 enrichment culture (dominated by SPV1), and conversely, the 3 spacers targeting SPV2 were from J14 enrichment culture (dominated by SPV2). However, according to the result in Figure 4a, SPV1-targeting spacers were enriched in J14 sample (dominated by SPV2), and conversely, SPV2-targeting spacers were enriched in J15 sample (dominated by SPV1). These data seem to be controversy, and need be confirmed or explained.

4) The title “Virus-borne mini-CRISPR arrays promote interviral conflicts and virus speciation” is eye-catching, but the related evidence in this paper is limited.

Minor points:

P3 Line84: This number is likely 40,704 according to Supplementary Data 1.

P3 Line115: “841 unique spacers” does not coincide with Figure 1d.

P4 Figure 1b: the meaning of letters H, R, Y, W and so on in the CRISPR repeat sequences, should better be specified in legend.

P4 Line137: ‘as in Figure 1’, Figure 1b?

P5 Line152: This speculation is arbitrary, are there any repeat experiments for the enrichment assay?.

P6 Lines 175-180: More information of the six most abundant spacers should be provided, including the sequence, PAM sequence, and importantly, abundance in each of the six samples. In addition, although here the authors indicated that “the result was inconsistent with an a priori expectation of negative correspondence between the frequency of spacers and the targeted viruses”. In some other places, the authors discussed that “the reciprocal CRISPR targeting by SPV1 and SPV2 strongly suggests that the virus-encoded mini-CRISPR arrays are involved in interviral conflicts and represent a distinct mechanism of superinfection exclusion, whereby a cell infected by one virus becomes resistant to the other virus. This possibility is consistent with the observation that J14 and J15 cultures contain exclusively SPV2 and SPV1, respectively” (P7 line 228-232), I am confused with these results.

P6 Line 172: Cite Figure 3a for this sentence (not for the next), and give more details for the “coincidence”. Explain how the P value is calculated?

P6 Line186: This inference is arbitrary. Actually, it could be verified by amplifying and sequencing the SPV1 or SPV2 intergenic region using specific primers and the enrichment sample.

P6 Lines 199-204: This part seemingly discussed the already existing spacers in SPV1 and SPV2 genomes, not from the HTS data. This information should be clearly given, also in Figure 3b and 4b, to avoid confusion.

P6 Line206: The Figure citation is incorrect, should be Figure 4b.

P6 Line209: This conclusion is a bit of arbitrary.

P7 Figure 3: Are the three mini-CRISPR arrays same as viral contigs 5, 6, 7 in Figure 4b? This information should be clearly given in legend.

P7 Line245: Citation of Figure S5 is missing here?

P8 Figure 4b: Add information for the pink and blue boxes.

P8 Line256: The definition of long and short CRISPR arrays seemed to be predicted from the CRISPRome data, which should be explained or specified here to avoid confusion.

P9 Line299: The design and usage of the primer pair G1 should be given.

P14 Supplementary Figure 4: The title is misleading. More explanation for this figure is needed, e.g. the values (seemingly they do not stand for the shared fraction), the color, and how this data support the similar initial spacer composition (as mentioned in P5 Line150).

Reviewer #2 (Remarks to the Author):

The paper studies the diversity of four CRISPR four loci found in Sulfolobales. They analyse two samples (J14 and J15) from Beepo (Japan) and their respective enrichments in genus specific media after 10 and 20 days. Sets of primers are used to amplify the mentioned loci from the surrounding areas and subsequently sequenced by next generation sequencing. They find more different spacer sequences (unique spacers or spacer clusters) than in all the previously known Sulfolobales sequences.

The authors report a concordance with shared unique spacers and geographical provenience of the isolate. Also study their evolution in enrichment media and found mini-arrays located in viruses.

The presence of CRISPR in prokaryotic viruses is not a novelty, but the characterization in Sulfolobales and the analysis of their dynamics is worth the read. I recommend publication, but some minor issues should be addressed as mentioned below.

General considerations.

I missed a brief description of the CRISPR-Cas systems. I think the CRISPR-Cas types or subtypes of the repeats should be mentioned if possible.

Also, I don't think it is appropriate to name the four CRISPR sequences "types". It is better to reserve this word for the CRISPR-Cas types. For instance, instead of "CRISPR type A" it is possible to use the nomenclature "CRISPR-A", or call them CRISPR consensuses (eg CRISPR consensus A).

Other comments:

-Lines 29-31: "The 25 million CRISPR spacers sequenced from a single sampling site dwarf the diversity of spacers from all available Sulfolobales isolates and display complex temporal dynamics"

This is the first time in the abstract that the word Sulfolobales is mentioned. Before that, it seems that the paper is about all CRISPR in an environmental sample. The restriction to some arrays in Sulfolobales should be made clearer. The reader at that point wonders why spacers are compared to the ones in Sulfolobales isolates.

-Lines 37-38: "present in the same population. Targeting by virus-borne spacers might represent a distinct mechanism of superinfection exclusion and appears to promote archaeal virus speciation"

I don't think that superinfection exclusion is the right expression. Superinfection exclusion (SIE) usually refers to the ability of a virus to prevent itself to infect the same cell several times. This case seems more related to intracitoplasmatic competition for the host between different strains of viruses. And certainly, this system would not impede re-infection with the same strain. Perhaps this indicates that there is not a SIE in the spacer containing virus compatible with a SIE in the targeted virus (in that example compatible SIE would imply that co-infection is avoided).

-Line 87. (48 spacers)

Better: (48 unique spacers)

-Line 88. "between the Beppu spacer set"

It should be made clearer at some point that the Beppu spacer set is the one reported in the present paper and not one previously described (eg in reference 35).

-Line 122. I would swap panels a and b, as b is more general than a. Current panel b would be a and vice versa.

-Line 137: "as in Figure 1"

Redundant as it is the figure 1 panel C, remove.

-Lines 144-147: "Despite possible biases introduced by PCR amplification, CRISPR spacers sequenced from the same replicon generally get similar representation in HTS reads"

In my experience it is not necessarily true that different arrays are amplified with similar efficiency. Perhaps PCR reactions with a mix of templates from different length arrays could be used to check this point for the different sets of primers used in this work.

Also, I don't know if this is compatible with what is shown in Supplementary Figure 8. I don't know if I have interpreted the figure correctly, but it seems that spacers from probably the same array (spacers belong to the same graph) have very different coverages (although the ratio of coverages between two samples for each spacer is similar). Does the coverage depend on the position of the spacer in the array?

“Therefore, the abundances of spacers show a multimodal distribution (Figure 2), likely reflecting the number of spacer-carrying Sulfolobales strains in the sample.”

Are there no spacers shared among different strains? If many spacer graphs cannot be resolved into one array probably there are; and although the distributions in Figure 2 are dependent on the proportion of different strains, each peak in a “discrete” distribution doesn’t necessarily represent a given strain. A peak with the highest representation could be formed by spacers present in a few strains.

-Lines 178-180: “This result was inconsistent with an a priori expectation of negative correspondence between the frequency of spacers and the targeted viruses.”

Although the presence of a virus implies sensitive hosts, I think it is to be expected that CRISPR targeting strains would be selected in a complex mixed culture -as are enrichments- where the targeted virus grows. The rest of the spacers may be present but not selected. This could even be happening if in the original culture there is only one host strain infected with the non-lytic virus, as some resistant strain could arise. Perhaps it can be expressed that this is against the general tendency in the abundance of spacers targeting either SPV1 or SPV2.

-Line 186: “spacers originated from mini-CRISPR arrays in SPV1 and SPV2 genomes”

I think that what is meant is “spacers were present in mini-CRISPR arrays from SPV1 and SPV2 genomes”. The origin of spacers implies a protospacer through acquisition. Also, were the arrays present in SPV1 and SPV2 or in related strains? Were the spacers self-targeting? The sentence as is may be misleading if the mini-arrays are in different strains of a similar virus.

-Line 192: “the SPV-encoded mini-CRISPR arrays are not associated with recognizable cas genes.”

Perhaps at the end of the sentence it could be added “in cis” or “in the viral genome”, as there are associated cas genes in the host genome. Other option is to change “associated” to “accompanied”.

-Lines 231-232: “the observation that J14 and J15 cultures contain exclusively SPV2 and SPV1, respectively (Supplementary Figure 1).”

The phrasing is confusing. A possible wording could be: “the observation that the cultures contain either SPV2 (J14) or SPV1 (J15), not both”

-Lines, 235-236. Is only SPV1 non-lytic? What about SPV2?

-Line 243. “, albeit without spacers,” Seems obvious for a stand-alone CRISPR sequence (in the MGE context alone it is not really a repetition as there is only one instance of the sequence). Remove.

-Line 249. Figure 4a. As in the other Figures (including supplementary) Sample J15 is depicted first I suggest putting J14 in the left and J15 in the right.

-Lines 245-246. “However, the stand-alone repeats were not preceded by recognizable leader sequences. Whether such repeats are competent targets for spacer integration is thus unclear.”

Consider the possibility that these stand-alone CRISPR sequences may serve another purpose such as CRISPR activity detection. Are they transcribed and in the same sense that in the CRISPR array? As authors surely know the CRISPR transcript is in most systems cleaved at the repeated sequence, and cleavage of an RNA could activate a response to CRISPR immunity in the MGE.

-Lines 223-224: “Identities between spacer and protospacers are indicated next to the protospacer bars” (In Figure 3)

I think it would be interesting to also indicate identity with the same strain harbouring the spacer. It would be interesting to know if PAM and seed sequences (when known) are correct.

-Lines 280-283. Are the described proportions of organisms at time $t=0$?

-Lines 359-367. (Prediction of mini-CRISPR arrays in the CRISPRome data)

The redaction of this section is a bit confusing. If the beginning of the section is about how to estimate the P-values that one spacer is alone in the array it should be specified before changing to mini-arrays with two spacers. In the case of mini-arrays with two spacers, only the P-value that a

given spacer is the first one in the array (there are no spacers before) is explained. I think this explanation is incomplete. Was the same done for the last spacer to verify it was the last? I'd rather say (starting from similar logic, ie that the probability of a given spacer to be first or last is 0.5) that the P-value of the two-spacer array being complete is approximately:

$$1 - (1 - 0.5^{(n-1)})^2$$

being n the number of read pairs with both spacers, and if none of both spacers has been read in any other pair.

That could also more easily being confirmed if the spacers are read with the delimiting sequences surrounding the array. Anyways, that doesn't change much the conclusion.

Reviewer #3 (Remarks to the Author):

Major comments

This is an interesting and potentially important paper showing strong evidence for inter-viral targeting by viral-encoded mini CRISPR arrays. Those mini-arrays have leader sequences and are highly active in new spacer acquisition and drive viral diversification, which is exciting indeed.

However, there are two key issues that need to be addressed. The first has to do with complete reliance on amplicon sequencing. The so called "CRISPRome" on which the analysis rests is based on amplicon sequencing, and it is well established that when amplifying gene fragments that differ in length, PCR will bias in favor of the smaller ones. Thus, if indeed "DNA fragments of 300–1000 bp in length were purified from the gel and sequenced on MiSeq", as described in the methods then there will be a bias in favor of the ~300 and against the ~1000 amplicons, not to mention that the short amplification time used may have already biased in favor of shorter arrays. Thus, the data are likely to be highly skewed in favor of mini arrays. Some of the consequences will still hold true, because some of the comparisons are of the same arrays across time, and so they will always be biased similarly. Still, most other statements will require additional validation either by direct quantification (DNA dot-blot hybridization), which is likely to be highly time-consuming or alternatively with shotgun metagenomics. Metagenomics, which while not going as deep in terms of sensitivity of spacer detection, should allow comparison of different arrays in a community in a relatively unbiased fashion and thus provide a reference for assessment of the amplification biases. A second issue is that is related to the first is that it is difficult to understand how the primers were designed, and how they also amplified the mini-arrays that are virus-encoded if they were based on chromosomal loci. Were the primers repeat based or leader based?

Minor comments

What do the green and red ORFs in Figure 3 encode?

Line 235 "Importantly, SPV1 is a non-lytic virus, which establishes a chronic infection and is released without killing its host (44). Therefore, the association between a non-lytic, CRISPR-bearing virus and the

host is beneficial to both parties and can thus be considered a form of symbiosis" - just because a virus is non-lytic does not imply that it has no fitness cost for the host, a chronic infection can reduce microbial fitness a great deal (for example infection with filamentous phages in bacteria can be highly deleterious to growth rate)

Supplementary table 2 "Schao" should be "S-Chao"

POINT-BY-POINT ANSWERS TO REVIEWERS' COMMENTS

REVIEWER 1

1) Are the type-specific primers (A, B, C, D) good enough to distinguish the four CRISPR types? Especially that the CRISPR repeat types A and D share the last few nucleotides. The evidences for this specificity should be provided before counting the spacers belonging to the arrays with the four repeat sequences (Figure 1).

RESPONSE: This is a valid inquiry. To avoid amplification of the same spacer with different primer pairs, we optimized primer sequences. Although the forward primers for type A and D CRISPR repeats are likely to be indiscriminative, the reverse primers are specific to repeats of types A and D, respectively.

To verify specificity of our primers for each CRISPR repeat type, we calculated the fraction of spacers that could be found between CRISPR repeats of different types, allowing 2 mismatches and end gaps/insertions. The result is presented in the figure below. For types A and D, we obtained 1.5% and 2% of intersection, i.e., 1.5% of spacers bordered by A-type CRISPR repeats (and amplified with A-specific primers) were also found in arrays bordered with D-type repeats. Importantly, this fraction was comparable to those for other repeat types. For instance, for types A and C, the intersection was 2.8% and 2.6%, suggesting that there is no preferential cross-amplification of spacers associated with A and D type repeats.

Figure 1. Intersection between spacer sets of different CRISPR repeats. The fraction of spacers from the CRISPR repeat type in a row matching with CRISPR repeat type in the column is indicated.

*The association of homologous spacers with different CRISPR repeat types can be explained by independent acquisition of the corresponding spacers from the same loci of the plasmid/viral genome by two distinct adaptation modules. This possibility was confirmed by analysis of spacer dataset from fully sequenced *Sulfolobus islandicus* genomes. With the same comparison parameters (2 mismatches and end gaps/insertions), we found 14 out of 552 (2.5%) spacers intersecting between A and D CRISPR repeats in fully sequenced genomes. This fraction closely matches that obtained in the analysis of the spacers in the Japanese Beppu dataset, and further suggests that cross-amplification of spacers associated with repeat types A and D is negligible.*

This information, along with the figure, is now provided as a new section in the Methods ("Evaluation of primer specificity").

2) The estimated PCR sizes are likely incorrect. As the theoretical size of a PCR product containing one spacer is the sum of $(24-31)*2$ (size of two repeats), 40 (the average size of spacer), and $33*2$ (two adaptors), which is around 165 bp. And this size will be around 235 bp for two-spacer containing PCR products. These sizes are out of the range (300-1000 bp) during PCR products purification (P9 Line301), but the authors implied that these products (Approximately 50% of our HTS sequencing reads include not solitary spacers but small fragments of CRISPR arrays with 2...) were the majority in the reads (P20 Line475).

RESPONSE: Thank you for pointing this out. There is indeed certain confusion. The excised gel fragments migrated in the 300-1000 bp range, as originally stated in the text (please, see the left panel in the figure). However, when, prior to sequencing, the same samples were analyzed using Bioanalyzer, a chip-based capillary electrophoresis machine, the fragments migrated at a lower molecular size, in the range of 100-400 bp (see the right panel in the figure below. The difference of migration in the agarose gel and capillary electrophoresis may be due to the low resolution of 1% agarose gel or overloading of the samples.

*Since the analysis of bioanalyzer is believed to be much more accurate, the actual sizes of the PCR products should be ranging from 100 to 300 bp. We have corrected this in the text. Besides, the HTS read length is 250 bp; thus, one read maximally contains 3 spacers: $5*24$ (5 repeats) + 3 spacers ($40*3$)=240 bp, which matches the sizes of our PCR products and the sequencing results (“Approximately 50% of our HTS sequencing reads include not solitary spacers but small fragments of CRISPR arrays with 2 or, more rarely, 3 spacers”).*

3) P6 Lines 175-180: The 3 most abundant spacers targeting SPV1 were sequenced from J15 enrichment culture (dominated by SPV1), and conversely, the 3 spacers targeting SPV2 were from J14 enrichment culture (dominated by SPV2). However, according to the result in Figure 4a, SPV1-targeting spacers were enriched in J14 sample (dominated by SPV2), and conversely, SPV2-targeting spacers were enriched in J15 sample (dominated by SPV1). These data seem to be controversy, and need be confirmed or explained.

RESPONSE: We apologize for the confusion. On page 6 (l 175-180, in the original manuscript), we report the initial observation that 3 spacers matching (100% identity) SPV1 were present in the sample dominated by SPV1 and, conversely, spacers matching SPV2 were sequenced from the SPV2-dominated sample. This highly unexpected observation prompted us to investigate the genomic context of the SPV-targeting spacers and subsequently led to the realization that the spacers are not carried by cellular CRISPR arrays, as originally presumed, but by viruses themselves. Accordingly, in the same paragraph (page 6, l 185-187), we clarify the observation by stating that “the 6 most abundant CRISPRome spacers originated from mini-CRISPR arrays in SPV1 and SPV2 genomes, rather than from the Sulfolobales genomes.” Knowing this, in Figure 4a (now 4c), we discriminate between the spacers carried by long, cellular CRISPR arrays and viral mini-arrays. Naturally, because J14 is dominated by SPV1, SPV2-targeting spacers carried by SPV1 are enriched in this sample. The opposite is true for sample J15: SPV2 is dominant and the spacers encoded by this virus are, accordingly, enriched.

We modified the sentence referring to Figure 4c in the following way: “To assess the effects of virus-mediated versus host-mediated CRISPR immunity against SPV1 and SPV2, we compared the number and abundance of the targeting spacers originating from the long host-borne CRISPR arrays and virus-borne mini-CRISPR arrays”. We also added the following explanation at the end of the paragraph: “ Thus, SPV1 and SPV2 came to dominance in the 20-days cultures of J14 and J15 samples, respectively, consistent with the enrichment of the SPV1-borne and SPV2-borne spacers/mini-arrays in the corresponding samples.”

We hope that the above explanation clears up the confusion.

4) The title “Virus-borne mini-CRISPR arrays promote interviral conflicts and virus speciation” is eye-catching, but the related evidence in this paper is limited.

RESPONSE: We agree that the evidence for virus speciation might be insufficient at this point and, thus, removed this part from the title.

P3 Line84: This number is likely 40,704 according to Supplementary Data 1.

RESPONSE: Thank you for pointing out this mistake. Yes, it should be 40,704 and not 40,705.

P3 Line115: “841 unique spacers” does not coincide with Figure 1d.

RESPONSE: 841 is the number of unique spacers independently identified in 6 analyzed samples. However, after the clustering of spacers from different samples, this number dropped to 521. In the revised manuscript we now state the latter number (521). Furthermore, we noticed that in the original Figure 1d the scale was restricted to 450 spacers. This is now also corrected.

P4 Figure 1b: the meaning of letters H, R, Y, W and so on in the CRISPR repeat sequences, should better be specified in legend.

RESPONSE: H,R,Y,W letters are IUPAC symbols used for combination of several nucleotides: H = A, C or T; Y = C or T; R = A or G; M = A or C; W = A or T; K = G or T. We added this information in the Figure legend, as suggested.

P4 Line137: ‘as in Figure 1’, Figure 1b?

RESPONSE: Corrected to “as in panel a”.

P5 Line152: This speculation is arbitrary, are there any repeat experiments for the enrichment assay?

RESPONSE: We removed the speculative statement. The sentence now reads: “Given that the strains from both samples were propagated under the same conditions, and initially displayed similar spacer composition (Supplementary Figure 4), we hypothesized that viruses present in enrichment cultures might be responsible for the differences in the growth dynamics of some of the strains.”

P6 Lines 175-180: More information of the six most abundant spacers should be provided, including the sequence, PAM sequence, and importantly, abundance in each of the six samples. In addition, although here the authors indicated that “the result was inconsistent with an a priori expectation of negative correspondence between the frequency of spacers and the targeted viruses”. In some other places, the authors discussed that “the reciprocal CRISPR targeting by SPV1 and SPV2 strongly suggests that the virus-encoded mini-CRISPR arrays are involved in interviral conflicts and represent a distinct mechanism of superinfection exclusion, whereby a cell infected by one virus becomes resistant to the other virus. This possibility is consistent with the observation that J14 and J15 cultures contain exclusively SPV2 and SPV1, respectively” (P7 line 228-232), I am confused with these results.

RESPONSE: In the revised manuscript, we included a new table within Supplementary data 2, which lists the characteristics of the six spacers, including their sequences, PAM sequences, identities to the target and self (orthologous locus), and abundance in each of the six samples. Please see our comment above regarding the unfortunate confusion. We hope that the provided explanations now clear it up.

P6 Line 172: Cite Figure 3a for this sentence (not for the next), and give more details for the “coincidence”. Explain how the P value is calculated?

RESPONSE: Figure 3a is now cited in the preceding sentence, as suggested. To calculate the P-value for the coincidence of spacer targeting and regions of divergence in SPV1 and SPV2 genomes, we aligned SPV1 and SPV2 genomes with ClustalW and manually verified the quality of nucleotide sequence alignment. Next, we calculated the number of spacers in our dataset that match fully conserved regions of SPV1-SPV2 alignment (22 spacers) and variable parts of the alignment (42 spacers). We simulated 10000 spacers from random positions of SPV1-SPV2 alignment and calculated the number of simulated spacers from conserved regions (5674 spacers) and variable regions of the alignment (4326 spacers). Chi square test was performed for observed (22 and 42) and expected (5674 and 4326) values; the resulting P-value was 0.006. This information is now included in the Methods section (“Protospacer analysis” subsection).

P6 Line186: This inference is arbitrary. Actually, it could be verified by amplifying and sequencing the SPV1 or SPV2 intergenic region using specific primers and the enrichment sample.

RESPONSE: As suggested, we performed PCR amplification of both mini-CRISPR arrays of SPV1 and SPV2 using as templates DNA from purified SPV1 virions as well as from the 20-days enrichment cultures J14 and J15 used for CRISPRome sequencing. The amplicons were sequenced and confirmed to correspond to the expected CRISPR loci. This information is now added to the text. See also new Supplementary figure 5.

P6 Lines 199-204: This part seemingly discussed the already existing spacers in SPV1 and SPV2 genomes, not from the HTS data. This information should be clearly given, also in Figure 3b and 4b, to avoid confusion.

RESPONSE: Indeed, here we describe spacers encoded within the SPV1- and SPV2-carried mini-arrays, rather than HTS data. The discovery of spacers in the SPV1 and SPV2 genomes based on the analysis of HTS data is described in the preceding paragraph:

“We found that the mini-CRISPR arrays including type A CRISPR repeats flanking the SPV-targeting spacers are encoded in intergenic regions of both SPV1 and SPV2 genomes. Thus, the 6 most abundant CRISPRome spacers originated from mini-CRISPR arrays in SPV1 and SPV2 genomes, rather than from the Sulfolobales genomes. The relative positions of the mini-CRISPR arrays containing 2 spacers in the SPV1 and SPV2 genomes were the same, but the corresponding spacers were different, implying active spacer turnover. These mini-CRISPR arrays are preceded by the promoter-containing leader sequences similar to those found in genomic Sulfolobus CRISPR arrays”.

Figure 3b schematically depicts the information provided on lines 199-204; namely, “two of the three spacers carried by SPV2 target SPV1, whereas only one of the three spacers carried by SPV1 targets SPV2”. We now added a pointer to Figure 3b. To illustrate that “loci orthologous to the regions targeted by spacers in the viruses carrying the spacers contain either point mutations or deletions, preventing self-targeting” we show in the newly included Supplementary data 2 the identity values of the SPV1 and SPV2 spacers to their corresponding targets and orthologous regions within the viruses carrying the spacers. We hope this is sufficient to clarify the confusion.

P6 Line206: The Figure citation is incorrect, should be Figure 4b.

RESPONSE: Thank you, this is now corrected.

P6 Line209: This conclusion is a bit of arbitrary.

RESPONSE: Identified candidates of mini-arrays are similar to mini-arrays in SPV in terms of the number of spacers (1-2 spacers) and provenance of spacers (the majority of spacers target SPV1 or SPV2). Moreover, 3 candidates of mini-arrays were found in the small viral contigs related to SPV1 or SPV2. We agree that the word “conclude” might be too strong for the evidence we have, so we replaced it with “suggest”.

P7 Figure 3: Are the three mini-CRISPR arrays same as viral contigs 5, 6, 7 in Figure 4b? This information should be clearly given in legend.

RESPONSE: Yes, they are the same. In the revised figure, the contigs are labeled and the information is added to the legend: “Mini-arrays found in SPV1, SPV2 and viral contigs 1-3 shown in Figure 3b are labeled on the left”.

P7 Line245: Citation of Figure S5 is missing here?

RESPONSE: We added citation of Figure S5.

P8 Figure 4b: Add information for the pink and blue boxes.

RESPONSE: We apologize for this oversight. Pink and blue boxes indicate two pairs of identical spacers found in mini-array candidates. We added this information to the Figure 4b legend.

P8 Line256: The definition of long and short CRISPR arrays seemed to be predicted from the CRISPRome data, which should be explained or specified here to avoid confusion.

RESPONSE: We apologize for the oversight. The reconstruction of the long CRISPR arrays is now introduced in the first section of the Results:

“Most amplicons including 2 or 3 spacer-repeat units and sequenced >100 times could be assembled into longer CRISPR arrays through identical spacers (Supplementary Text). The longest assembled

CRISPR arrays were 131, 66, 139 and 119 for spacers associated with the CRISPR-A, -B, -C and -D repeats, respectively.”

P9 Line299: The design and usage of the primer pair G1 should be given.

RESPONSE: Primers G1 were designed for CRISPR repeat sequence “ATTACTTTTCTCTTATGAGACTAGTAC” found in a single CRISPR array with only 3 spacers in S. tokodaii genome. The latter CRISPR array is not localized near cas genes in S. tokodaii. No PCR product was obtained with G1 primers from samples used in this study. This information is now specified in Materials and Methods.

P14 Supplementary Figure 4: The title is misleading. More explanation for this figure is needed, e.g. the values (seemingly they do not stand for the shared fraction), the color, and how this data support the similar initial spacer composition (as mentioned in P5 Line150).

RESPONSE: We added legend for the color and updated title and description of Figure S4.

REVIEWER 2

I missed a brief description of the CRISPR-Cas systems. I think the CRISPR-Cas types or subtypes of the repeats should be mentioned if possible.

RESPONSE: Thank you for the suggestion. A brief paragraph describing CRISPR-Cas systems and their classification is now added.

Also, I don't think it is appropriate to name the four CRISPR sequences “types”. It is better to reserve this word for the CRISPR-Cas types. For instance, instead of “CRISPR type A” it is possible to use the nomenclature “CRISPR-A”, or call them CRISPR consensus (eg CRISPR consensus A).

RESPONSE: In the revised manuscript, we used CRISPR-A, -B, etc. nomenclature.

-Lines 29-31: “The 25 million CRISPR spacers sequenced from a single sampling site dwarf the diversity of spacers from all available Sulfolobales isolates and display complex temporal dynamics”
This is the first time in the abstract that the word Sulfolobales is mentioned. Before that, it seems that the paper is about all CRISPR in an environmental sample. The restriction to some arrays in Sulfolobales should be made clearer. The reader at that point wonders why spacers are compared to the ones in Sulfolobales isolates.

*RESPONSE: The Sulfolobales order has now been introduced earlier:
“Here, we performed deep sequencing of the CRISPRome—all spacers contained in a microbiome—associated with hyperthermophilic archaea of the order Sulfolobales recovered directly from an environmental sample and from enrichment cultures established in the laboratory.”*

Lines 37-38: “present in the same population. Targeting by virus-borne spacers might represent a distinct mechanism of superinfection exclusion and appears to promote archaeal virus speciation”
I don't think that superinfection exclusion is the right expression. Superinfection exclusion (SIE) usually refers to the ability of a virus to prevent itself to infect the same cell several times. This case seems more related to intracitoplasmatic competition for the host between different strains of viruses. And certainly, this system would not impede re-infection with the same strain. Perhaps this indicates that there is not a SIE in the spacer containing virus compatible with a SIE in the targeted virus (in that example compatible SIE would imply that co-infection is avoided).

RESPONSE: Thank you for this comment, which helps us to clarify what we meant. The term superinfection exclusion can be used to describe protection against closely related viruses (homotypic), but also against distinct/distantly related viruses (heterotypic). In the literature, these terms are used for both prokaryotic and eukaryotic viruses (e.g., PMID: 28067906 and 3041035). In the case of SPV1 and SPV2, we meant the heterotypic mode of superinfection exclusion and this is made clear in the revised manuscript:

*“Targeting by these virus-borne spacers represents a distinct mechanism of **heterotypic** superinfection exclusion...”*

-Line 87. (48 spacers)

Better: (48 unique spacers)

RESPONSE: Corrected.

-Line 88. “between the Beppu spacer set”

It should be made clearer at some point that the Beppu spacer set is the one reported in the present paper and not one previously described (eg in reference 35).

RESPONSE: The sentence has been amended:

“The largest intersection (48 unique spacers) was found between the obtained spacer set from Beppu and spacers from the previously sequenced Sulfolobales strains isolated in Japan”.

-Line 122. I would swap panels a and b, as b is more general than a. Current panel b would be a and vice versa.

RESPONSE: The figure has been modified as suggested.

-Line 137: “as in Figure 1”

Redundant as it is the figure 1 panel C, remove.

RESPONSE: The figure and legend have been modified.

-Lines 144-147: “Despite possible biases introduced by PCR amplification, CRISPR spacers sequenced from the same replicon generally get similar representation in HTS reads”

In my experience it is not necessarily true that different arrays are amplified with similar efficiency.

Perhaps PCR reactions with a mix of templates from different length arrays could be used to check this point for the different sets of primers used in this work. Also, I don’t know if this is compatible with what is shown in Supplementary Figure 8. I don’t know if I have interpreted the figure correctly, but it seems that spacers from probably the same array (spacers belong to the same graph) have very different coverages (although the ratio of coverages between two samples for each spacer is similar). Does the coverage depend on the position of the spacer in the array?

RESPONSE: We tested the CRISPRome approach by using primers specific to E. coli I-E CRISPR repeat to amplify spacers from E. coli MG1655 K12 lab strain (PMID: 27997045) or a mixture of E. coli strains (not published). In E.coli, we observed reduced amplification efficiency for the first spacer in the CRISPR array and amplification of shorter CRISPR array (6 spacers) was on average more efficient than longer CRISPR array (12 spacers). Given that the average size of the CRISPR arrays in Sulfolobales is 70, we consider that the bias of the first spacer to be negligible. From the model experiment, we estimated that variation in spacer abundances from one array was no greater than 5-fold. To minimize the visual effect of this variation, we use the logarithmic scale in Figure 2.

“Therefore, the abundances of spacers show a multimodal distribution (Figure 2), likely reflecting the number of spacer-carrying Sulfolobales strains in the sample.”

Are there no spacers shared among different strains? If many spacer graphs cannot be resolved into one

array probably there are; and although the distributions in Figure 2 are dependent on the proportion of different strains, each peak in a “discrete” distribution doesn’t necessarily represent a given strain. A peak with the highest representation could be formed by spacers present in a few strains.

RESPONSE: Multimodal distribution was found only in cultivated samples, which contained a decreased number of Sulfolobales strains surviving in the artificial growth conditions (as can be estimated from the total number of spacers in our samples and the average number of spacers in Sulfolobales genomes). However, the possibility that the highly represented groups include spacers from more than one strain cannot be formally excluded. This is now clearly stated in the revised text.

-Lines 178-180: “This result was inconsistent with an a priori expectation of negative correspondence between the frequency of spacers and the targeted viruses.”

Although the presence of a virus implies sensitive hosts, I think it is to be expected that CRISPR targeting strains would be selected in a complex mixed culture -as are enrichments- where the targeted virus grows. The rest of the spacers may be present but not selected. This could even be happening if in the original culture there is only one host strain infected with the non-lytic virus, as some resistant strain could arise. Perhaps it can be expressed that this is against the general tendency in the abundance of spacers targeting either SPV1 or SPV2.

RESPONSE: We agree with the reviewer. We decided to remove the highlighted sentence.

-Line 186: “spacers originated from mini-CRISPR arrays in SPV1 and SPV2 genomes”

I think that what is meant is “spacers were present in mini-CRISPR arrays from SPV1 and SPV2 genomes”. The origin of spacers implies a protospacer through acquisition. Also, were the arrays present in SPV1 and SPV2 or in related strains? Were the spacers self-targeting? The sentence as is may be misleading if the mini-arrays are in different strains of a similar virus.

RESPONSE: What we meant is that the 6 spacers in our CRISPRome data, in contrast to the majority of CRISPRome spacers, were amplified and sequenced not from Sulfolobus genomes, but from the viral genomes. We rephrased this sentence in the text:

“Thus, the 6 most abundant CRISPRome spacers were amplified from mini-CRISPR arrays in SPV1 and SPV2 genomes, rather than from the Sulfolobales genomes”.

-Line 192: “the SPV-encoded mini-CRISPR arrays are not associated with recognizable cas genes.”

Perhaps at the end of the sentence it could be added “in cis” or “in the viral genome”, as there are associated cas genes in the host genome. Other option is to change “associated” to “accompanied”.

RESPONSE: We agree, the wording was not accurate. In the manuscript text “associated” was replaced with “accompanied”, as suggested.

-Lines 231-232: “the observation that J14 and J15 cultures contain exclusively SPV2 and SPV1, respectively (Supplementary Figure 1).”

The phrasing is confusing. A possible wording could be: “the observation that the cultures contain either SPV2 (J14) or SPV1 (J15), not both”

RESPONSE: Thank you, the text has been changed as suggested.

-Lines, 235-236. Is only SPV1 non-lytic? What about SPV2?

RESPONSE: Unfortunately, only SPV1 is available in pure culture and infection cycle was studied only with this virus. Hence, we mention only SPV1 in that particular sentence, although given that the two viruses are 92% identical at the nucleotide level, the mechanism of egress is highly likely to be the same for SPV1 and SPV2.

-Line 243. “,albeit without spacers,” Seems obvious for a stand-alone CRISPR sequence (in the MGE context alone it is not really a repetition as there is only one instance of the sequence). Remove.

RESPONSE: Removed, as suggested.

-Line 249. Figure 4a. As in the other Figures (including supplementary) Sample J15 is depicted first I suggest putting J14 in the left and J15 in the right.

RESPONSE: Thank you for the suggestion. However, we prefer showing J15 first, because it contains the most diverse, non-cultivated sample “J15-0 days”. We corrected the sample order in Figure 4 to match other Figures.

-Lines 245-246. “However, the stand-alone repeats were not preceded by recognizable leader sequences. Whether such repeats are competent targets for spacer integration is thus unclear.” Consider the possibility that these stand-alone CRISPR sequences may serve another purpose such as CRISPR activity detection. Are they transcribed and in the same sense that in the CRISPR array? As authors surely know the CRISPR transcript is in most systems cleaved at the repeated sequence, and cleavage of an RNA could activate a response to CRISPR immunity in the MGE.

RESPONSE: It is an interesting possibility that standalone CRISPR repeats in viruses can activate CRISPR immunity of the host. As shown in the Figure 5b, stand-alone CRISPR repeats can be found in multiple Sulfolobales viruses, sometimes even in multiple copies. However, stand-alone repeats in SPV viruses do not have recognizable promoter sequences nearby on the forward or the reverse strand (Figure 4). Thus, their transcription appears unlikely. The genomic context of these repeats is not conserved in SPV1 and SPV2. Notably, the SPV2 genome contains a disrupted transposase gene next to the stand-alone repeat. In principle, the transcription of the SPV2 stand-alone CRISPR repeat could start from the transposase promoter located 400 bp upstream in the forward strand. However, the standalone CRISPR repeat would be transcribed in the opposite direction compared to the host pre-crRNA, and thus would not be recognized by cas6 protein for the cleavage. Thus, the role of stand-alone repeats remains enigmatic and we prefer refraining from suggesting their potential functions.

-Lines 223-224: “Identities between spacer and protospacers are indicated next to the protospacer bars” (In Figure 3)

I think it would be interesting to also indicate identity with the same strain harbouring the spacer. It would be interesting to know if PAM and seed sequences (when known) are correct.

RESPONSE: To answer these questions we added Supplementary Data file 2 with an additional information about the spacers from SPV mini-arrays, including abundance in different samples, sequence of the spacers, identity to target and self-targeting identity, PAM in the target and self-targeting PAM. Importantly, type III CRISPR-Cas interference complexes of Sulfolobus do not require the PAM sequence for the RNA and DNA cleavage. Thus, absence of canonic PAM sequence (CCN) will not prevent from type III targeting.

-Lines 280-283. Are the described proportions of organisms at time $t=0$?

RESPONSE: The described proportions are from 20-days cultivated samples. We added this information in the text of the manuscript.

-Lines 359-367. (Prediction of mini-CRISPR arrays in the CRISPRome data)

The redaction of this section is a bit confusing. If the beginning of the section is about how to estimate the P-values that one spacer is alone in the array it should be specified before changing to mini-arrays with two spacers. In the case of mini-arrays with two spacers, only the P-value that a given spacer is the first one in the array (there are no spacers before) is explained. I think this explanation is incomplete. Was the same done for the last spacer to verify it was the last? I'd rather say (starting from similar logic,

ie that the probability of a given spacer to be first or last is 0.5) that the P-value of the two-spacer array being complete is approximately:

$1 - (1 - 0.5^{(n-1)})^2$ being n the number of read pairs with both spacers, and if none of both spacers has been read in any other pair.

That could also more easily being confirmed if the spacers are read with the delimiting sequences surrounding the array. Anyways, that doesn't change much the conclusion.

RESPONSE: Thank you for the suggestion. We edited this section according to your comments. To avoid confusion, in the revised section, we describe estimation of the P-values for mini-array candidates with one and two spacers in separate paragraphs. Thank you for the formula for two-spacer array P-value calculation; we included it with comments in the modified text.

It would be indeed possible to define the borders of CRISPR arrays based on surrounding sequences. However, it is challenging to amplify and sequence these non-conserved delimiting sequences for novel CRISPR arrays.

REVIEWER 3

This is an interesting and potentially important paper showing strong evidence for inter-viral targeting by viral-encoded mini CRISPR arrays. Those mini-arrays have leader sequences and are highly active in new spacer acquisition and drive viral diversification, which is exciting indeed.

RESPONSE: Thank you for the constructive comments and positive assessment of our work.

However, there are two key issues that need to be addressed.

The first has to do with complete reliance on amplicon sequencing. The so called "CRISPRome" on which the analysis rests is based on amplicon sequencing, and it is well established that when amplifying gene fragments that differ in length, PCR will bias in favor of the smaller ones. Thus, if indeed "DNA fragments of 300–1000 bp in length were purified from the gel and sequenced on MiSeq", as described in the methods then there will be a bias in favor of the ~300 and against the ~1000 amplicons, not to mention that the short amplification time used may have already biased in favor of shorter arrays. Thus, the data are likely to be highly skewed in favor of mini arrays. Some of the consequences will still hold true, because some of the comparisons are of the same arrays across time, and so they will always be biased similarly. Still, most other statements will require additional validation either by direct quantification (DNA dot-blot hybridization), which is likely to be highly time-consuming or alternatively with shot-gun metagenomics. Metagenomics, which while not going as deep in terms of sensitivity of spacer detection, should allow comparison of different arrays in a community in a relatively unbiased fashion and thus provide a reference for assessment of the amplification biases.

RESPONSE: There appears to be slight misunderstanding regarding the design of the PCR amplification used. The designed primers were specific to members of the order Sulfolobales (based on sequences available in the GenBank) and were complementary to the CRISPR repeats, with the forward and reverse primers partially overlapping (see the figure below).

The primers could anneal at multiple sites in the CRISPR array; hence, the longer the array the more sites at which both forward and reverse primers can anneal. Consequently, amplification products from the cellular CRISPR arrays which, on average, contain >100 repeat-spacer units versus 2-3 in the viral mini-

arrays, are expected to be more abundantly represented. It is true that shorter products will be selected over longer ones, but the cellular arrays provide overwhelmingly more opportunities to produce short amplicons of cellular CRISPR spacers than the mini-arrays. Thus, there is no reason why the data would be skewed in favor of mini-arrays.

A second issue is that is related to the first is that it is difficult to understand how the primers were designed, and how they also amplified the mini-arrays that are virus-encoded if they were based on chromosomal loci. Were the primers repeat based or leader based?

RESPONSE: We apologize for not being clear in the original description. In the revised Methods section we added the following explanation:

“The forward and reverse primers in each pair were complementary to the CRISPR repeats, with the forward and reverse primers partially overlapping in their 5’-termini.”

The viral mini-arrays were amplified because their repeat sequences are nearly identical to those of the host (Figure 4).

Minor comments

What do the green and red ORFs in Figure 3 encode?

RESPONSE: In the revised version of the figure, we removed the coloring which, we agree, was somewhat confusing.

Line 235 "Importantly, SPV1 is a non-lytic virus, which establishes a chronic infection and is released without killing its host (44). Therefore, the association between a non-lytic, CRISPR-bearing virus and the host is beneficial to both parties and can thus be considered a form of symbiosis" - just because a virus is non-lytic does not imply that it has no fitness cost for the host, a chronic infection can reduce microbial fitness a great deal (for example infection with filamentous phages in bacteria can be highly deleterious to growth rate).

RESPONSE: We do not argue that there is no fitness cost. We merely suggest that the association between the virus and the host can be considered a form of symbiosis. Indeed, given that SPV viruses provide immunity in the form of CRISPR spacers targeting other viruses and mobile genetic elements, the host benefits from the CRISPR array carried by the virus (and virus, obviously, benefits from the host), which fits the definition of symbiotic relationship.

Supplementary table 2 "Schao" should be "S-Chao"

RESPONSE: Thank you, we corrected this mistake.

Reviewers' comments:

Reviewer #1 (Remarks to the Author):

The questions I have pointed out have been carefully revised or answered and I now agree to publish the paper.

Reviewer #2 (Remarks to the Author):

In the manuscript, Medvedeva et al. study CRISPR sequences of Sulfolobales present in two environmental samples (J15 and J14) from Beppu (Japan). The study also includes enrichment cultures. They use PCR amplification with primers specific to the four known CRISPR consensus (which they identify as A, B, C, and D) followed by high throughput sequencing and bioinformatic analysis.

More than 40,000 new spacer sequences are found, which can depict an accurate picture of underlying processes. They find that the different CRISPR have also different target preference. Some regions have so many targets that they can even be assembled from spacer sequences. Spacers were also useful to predict integrated mobile genetic elements.

Of special relevance are small CRISPR arrays (one to three repeats) present in viruses. This small arrays with spacers can use the associated Cas proteins coded in the host genome. Remarkably, in the present case, the small arrays include a leader sequence. The arrays are present in the two main viruses infecting Sulfolobus (SPV1 and SPV2), when in SPV1 they target SPV2 and vice versa. This could explain samples are dominated by only one of those viruses. Also, a few strains of viruses related carry related mini arrays.

The manuscript has improved a lot in the revised version. Based on the interesting dynamics and the importance of the involved species, I recommend publication.

I also have a few minor comments.

-Lines 271-273: "Importantly, SPV1 is a non-lytic virus, which establishes a chronic infection and is released without killing its host. Therefore, the association between a non-lytic, CRISPR-bearing virus and the host is beneficial to both parties and can thus be considered a form of symbiosis."

I don't think it is demonstrated that the relationship is necessarily beneficial to both parties. At least other options should not be excluded. Are the other viruses lytic for sure? Would it be beneficial if other viruses were not around? Perhaps the text could be changed from "beneficial" to "potentially beneficial".

-Line 333: "S15 (20 days) and S14 (20 days) enrichment cultures". I suppose the authors mean 'J15' and 'J14'.

-Lines 416-423: There is a reason why I used "N-1" instead of the "N" in your version of the formula. There is a 100% probability that a two-spacer read has a left and a right spacer, not 50% for each of them. Therefore, the confirmation that the given left or right spacer is at the extreme of the array starts with the second two-spacer read.

-Figure 5b: I suggest the inclusion of the number of arrays in the table.

-Figure 5c: Mini-arrays targeting the dominant virus seem to decrease in J15 after enrichment while the opposite is true for J14. If this is not an artefact (see below) I think this should be mentioned in the text. Also, suggest a possible explanation; e.g. Is it possible that SPV2 kills hosts and there is a bigger pool of uninfected cells?

You obtained a disproportionate amount of reads from J14 after 20 days of enrichment. Is the number of reads proportional to the concentration of CRISPR in the sample? (i.e. there is a much bigger proportion of CRISPR DNA in the sample compared to total DNA or per ml) If not, can the data from different samples be normalized so that a better comparison is possible?

Reviewer #3 (Remarks to the Author):

I think that the authors have addressed all points raised by the other referees and I well.

Lines 271-273: “Importantly, SPV1 is a non-lytic virus, which establishes a chronic infection and is released without killing its host. Therefore, the association between a non-lytic, CRISPR-bearing virus and the host is beneficial to both parties and can thus be considered a form of symbiosis.” I don’t think it is demonstrated that the relationship is necessarily beneficial to both parties. At least other options should not be excluded. Are the other viruses lytic for sure? Would it be beneficial if other viruses were not around? Perhaps the text could be changed from “beneficial” to “potentially beneficial”.

RESPONSE: With a limited available diversity of spacers from mini-arrays (26 spacers) we could not find any matches to genomes of lytic viruses. However, some of the spacers from mini-arrays matched other, non-SPV-like mobile genetic elements, including pRN1-like plasmid and temperate virus SSV17. Therefore, mini-arrays expressed from the genome of a non-lytic virus are highly likely to provide the same benefits as the host encoded CRISPR-Cas system. However, we agree that the statement was too strong without direct evidence and rephrased it as suggested.

-Line 333:” S15 (20 days) and S14 (20 days) enrichment cultures”. I suppose the authors mean ‘J15’ and ‘J14’.

RESPONSE: Thank you for noticing these typos. We corrected the text.

-Lines 416-423: There is a reason why I used “N-1” instead of the “N” in your version of the formula. There is a 100% probability that a two-spacer read has a left and a right spacer, not 50% for each of them. Therefore, the confirmation that the given left or right spacer is at the extreme of the array starts with the second two-spacer read.

RESPONSE: Thank you for this remark; we modified the formula in the revised manuscript and added a comment: “If the spacer was sequenced in the pair N times, 1 pair defines spacer as a first, and remaining (N-1) pairs are used to estimate the probability 0.5^{N-1} to observe spacer only as a first spacer in the pair.”

-Figure 5b: I suggest the inclusion of the number of arrays in the table.

RESPONSE: Modified as suggested.

-Figure 5c: Mini-arrays targeting the dominant virus seem to decrease in J15 after enrichment while the opposite is true for J14. If this is not an artefact (see below) I think this should be mentioned in the text. Also, suggest a possible explanation; e.g. Is it possible that SPV2 kills hosts and there is a bigger pool of uninfected cells?

RESPONSE: The dynamics in the J15 sample is indeed interesting: In the initial J15-0 days sample SPV2 was more abundant than SPV1 virus, which unexpectedly turned into SPV1 dominance in the enrichment samples. There could be at least 3 non-mutually exclusive explanations: 1) cells infected with SPV2 in the initial sample were less viable and were outcompeted by noninfected cells susceptible to SPV1; 2) Sulfolobus hosts were slightly better protected from SPV2 than from SPV1 (judging from the number of SPV2 targeting spacers shown in Figure 5c), leading to more efficient SPV1 proliferation; 3) other factors in the complex microbial community of the original sample selected for SPV2 proliferation.

You obtained a disproportionate amount of reads from J14 after 20 days of enrichment. Is the number of reads proportional to the concentration of CRISPR in the sample? (i.e. there is a much bigger proportion of CRISPR DNA in the sample compared to total DNA or per ml) If not, can the data from different samples be normalized so that a better comparison is possible?

RESPONSE: The number of reads in each sample is proportional to the number of CRISPR spacers in the sample, but is limited by the number of primers used in the amplification reaction. Notably, the

“concentration” of CRISPR spacers in the SPV genome is comparable to that in the host Sulfolobus genome:

in SPV: 3 spacers per 20kb genome = 0.15 spacers per kb;

in Sulfolobus: ~0.064 spacers per kb (calculated for 23 Sulfolobales genomes with type A CRISPR repeat).

During PCR amplification, CRISPR repeats of mini-arrays of SPV and long arrays of Sulfolobus compete for the same primers. As a result, the proliferation of SPV in 20 days samples might decrease the amount of reads from Sulfolobus long arrays. However, the ratio between total number of spacers from long arrays and mini-arrays or between SPV1 mini-arrays and SPV2 mini-arrays will not be affected and can be compared between different samples in Figure 5c. We modified the text by removing the conclusion of SPV predation of the host, which was based on the absolute numbers.